# Advances in the Non-Operative Management of Multidirectional Instability of the Glenohumeral Joint

**DOI:** 10.3390/jcm11175140

**Published:** 2022-08-31

**Authors:** Lyn Watson, Tania Pizzari, Simon Balster, Ross Lenssen, Sarah Ann Warby

**Affiliations:** 1Melbourne Shoulder Group, 305 High Street, Prahran, VIC 3181, Australia; 2Department of Physiotherapy, Podiatry, Prosthetics and Orthotics, La Trobe University, Corner of Kingsbury Drive and Plenty Road Bundoora, Bundoora, VIC 2080, Australia; 3Mill Park Physiotherapy, 22/1 Danaher Dr, South Morang, VIC 3752, Australia

**Keywords:** multidirectional instability, shoulder, rehabilitation, scapula, motor control, classification

## Abstract

Multidirectional instability (MDI) of the glenohumeral joint refers to symptomatic subluxations or dislocations in more than one direction. The aetiology of MDI is multifactorial, which makes the classification of this condition challenging. A shoulder rehabilitation program is the initial recommended treatment for MDI, however available rehabilitation programs have varying levels of evidence to support their effectiveness. In 2016, we published the details of an evidence-based program for MDI that has been evaluated for efficacy in two single-group studies and a randomised controlled trial. In 2017, we published a clinical commentary on the aetiology, classification, and treatment of this condition. The aim of this paper is to provide an update on the components of these publications with a particular focus on new advances in the non-operative management of this condition.

## 1. Introduction

While there are discrepancies in the explicit definition, most authors agree that multidirectional instability (MDI) is the symptomatic subluxation and/or dislocation of the glenohumeral joint in the inferior as well as anterior and/or posterior directions [1,2,3,4]. The glenohumeral joint subluxations or dislocations that occur in MDI cause pain and disability in sports, occupations, and activities of daily living. 

Understanding the factors contributing to MDI is paramount for the selection of the most appropriate treatment strategies. Despite the inconsistencies in the definition, classification, and diagnostic criteria of MDI, there is a general consensus across the literature on the typical aetiology, presentation, and treatment of this condition. 

In 2016, we published the details of an evidence-based program for MDI [5,6] that has been evaluated for efficacy in two single-group studies [7,8] and a randomised controlled trial [9]. In 2017, we published a clinical commentary on the aetiology, classification, and treatment of this condition [10]. The aim of this paper is to provide an update on the components of these publications with a particular focus on new advances in the non-operative management of this condition.

## 2. Aetiology

### 2.1. Passive Factors 

Excessive shoulder capsular laxity is a key predisposing factor in [11,12,13]. Patients with MDI have significantly more voluminous joint capsules [11] and defective rotator intervals [14,15] when compared to controls. This reduces passive restraint to glenohumeral joint motion, particularly at end range [16]. The higher rates of collagen turnover in people with MDI [12,13,17] also reduce the tensile strength of the tissue [18]. Capsular laxity may be congenital [13,19], and some authors have suggested it may be acquired due to repetitive microtrauma such as swimming or throwing [13,19,20], although this theory has yet to be empirically tested. It is important to distinguish laxity from instability. Hyperlaxity is an increase in joint translation due to the elongation of soft tissue structures, which is controlled by the sensorimotor system and is asymptomatic. Instability occurs when translation becomes uncontrolled and symptomatic [19,21]. Although laxity alone is non-pathological, it may predispose people to instability in the presence of aberrant motor patterning and/or microtrauma [22]. 

Generalised ligament laxity (GLL) may be present in some MDI patients although it is not always a typical finding [9,12]. GLL is part of the normal anatomical spectrum [19] and does not necessarily lead to the development of instability [23]. Extreme forms of congenital hyperlaxity, such as Ehlers–Danlos and Marfan’s Syndrome may be present in a small number of cases of MDI and are associated with other systemic disorders such as congenital heart defects, hormonal imbalances, digestive problems, and subluxations of other synovial joints [13,22,24,25]. 

Glenoid retroversion (glenoid facing more posteriorly) and glenoid hypoplasia (underdevelopment of the inferior glenoid ossification centre) are common in MDI [26,27] and reduce the small bony contact between the humeral head and glenoid. Reductions in passive restraint of the glenohumeral joint may predispose people to MDI in the presence of poor muscle control and/or exposure to micro-trauma [25].

### 2.2. Scapula 

Patients with MDI typically display a downward rotation of their scapulae at rest and a lack of upward rotation through arm elevation [8,28,29,30]. This reduces the relative contact area of the humeral head on the glenoid, which contributes to the quintessential sulcus sign (inferior instability) seen at rest in MDI, and pathological humeral head translations through range [31,32,33,34]. A reduction in bony congruence between the glenoid and the humeral head places additional strain on the already lax surrounding passive structures and inhibits the optimal length–tension relationship of the rotator cuff [12]; all of which affect the concavity compression mechanism of the humeral head on the glenoid. Aberrant scapular motor patterns are often referred to as scapular “dyskinesis”. It is unknown if aberrant scapular motor patterning is a cause or effect of MDI [35], although a lack of upward rotation is a risk factor for developing shoulder pain [36].

### 2.3. Altered Muscle Control

Electromyography (EMG) studies have shown patients with MDI have altered scapulo-thoracic, rotator cuff and deltoid function compared to people with stable shoulders [8,30,37], although the altered motor patterns are not consistent between studies [28]. The inconsistencies are likely due to the varied presentations of the pathology and the different movement strategies patients use to stabilize their shoulders. Patients with MDI display reduced variability of movement compared to controls [8]. Having fewer available recruitment strategies when performing certain tasks may overload soft tissue structures [38]. 

Motor alterations may develop in response to the inability of the passive structures to optimally limit and control motion in the glenohumeral joint or could be a response to experiencing pain. The presence of pain causes functional reorganization of muscle synergies (i.e., motor patterning), decreases motor unit discharge rates and reduces maximal voluntary contractions [39]. Altered muscle control in MDI might present as a secondary adaptation in response to microtrauma-induced pain or may be a primary phenomenon [35]. 

### 2.4. Altered Proprioception and Central Function

Patients with MDI have reduced proprioception, or joint position, sense when compared to controls [40,41,42]. Excessive laxity associated with microtrauma may damage the neural receptors in the shoulder joint capsule, disrupting afferent signalling to the central nervous system [35,43]. Consequently, efferent neuromuscular control of the scapular and humeral muscles is altered [44]. Proprioception deficiencies result in altered timing and amplitude of muscle contraction, increased movement at a joint segment, and an inability to correct motion errors for placing distal segments [40]; all of which affect joint function. Surgery to tighten the capsule can restore proprioception [41] and joint kinematics [45] to levels near normal, suggesting that capsular laxity may be a primary phenomenon in the development of poor proprioception in MDI. 

## 3. Classification 

There is no universally accepted system for classifying MDI [2,46]. In a previous publication [10], we described the strengths and limitations of several shoulder classification systems and the effect that their variable criteria may have on the number of patients diagnosed with MDI in a shoulder instability population [46]. The true incidence of MDI is unknown partly for this reason. 

Despite these present challenges, MDI experts agree that aetiology is an important consideration as it assists in the selection of the most appropriate treatment option [2,46,47,48,49]. A recent classification system developed in a large Delphi study for posterior shoulder instability [48] can be easily applied to MDI. The classification includes three subgroups: traumatic, micro-traumatic and atraumatic instability. While these sub-groups are headed by aetiology, they relate to a collection of associated factors that contribute to the typical presentation of that sub-group of instability. The classical MDI presentation typically falls in the atraumatic sub-group, with no seemingly obvious mechanism of injury or cause of onset, congenital hypermobility, up to three directions of instability, poor motor patterning, and a lack of structural lesions [12,35,48]. Evidence supports that these patients have optimal outcomes with rehabilitation [8,9,13,50,51,52]. 

MDI patients can also fall into the micro-traumatic group [53]. Often, patients who have a background of shoulder laxity that was previously asymptomatic and well controlled, start to experience symptoms in the inferior and anterior and/or posterior directions when motor control is lost in association with micro-trauma and pain [22,32]. These acquired MDI types are more likely to have instability in two directions [20]. This group has a larger variation in their associated factors; however, they have an increased incidence of acquired structural lesions due to the repetitive microtrauma imposed on their shoulder [32,48]. Management typically involves 6 months of evidence-based rehabilitation, and if this fails, surgery may be considered [48,54]. Clinicians should be aware that while these sub-groups and associated factors are typical, they are not completely rigid. For example, a patient with atraumatic MDI can fall over and sustain a traumatic structural lesion [55], which is likely to alter management. 

Understanding the associated factors that contribute to a patient’s presentation relies on a thorough clinical assessment, and then treatment can address all identified factors for optimal patient outcomes. 

## 4. Clinical Presentation

### 4.1. History

The severity of MDI can vary. In milder cases, patients may report vague shoulder pain, reduced strength, and/or altered sporting performance [19] without the perception of instability [53]. Conversely, others may present with violent subluxations with basic shoulder movements, apprehension and pain, and severely impacted activities of daily living [55]. Patients presenting with MDI are typically between 10 and 35 years old [24], however shoulder instability (traumatic, micro-traumatic and atraumatic) can still present beyond this range [56,57]. 

There is often a predominant direction of instability [13] with posterior/inferior reported to be the most common [5,9,55]. By carefully listening to the patient’s aggravating factors, the clinician can often identify the primary and secondary directions of instability [3,19,58] as subjective reports of the position of aggravation correlates well with objective instability tests [59]. As inferior instability is the quintessential mark of MDI, typically patients will report symptoms carrying a heavy bag by their side or a backpack on the affected side [32]. Patients who report symptoms primarily with combinations of flexion, horizontal flexion and adduction (e.g., taking off a tight top by crossing the arms across the chest, driving, pushing a heavy door, push-ups) often have a predominance of posterior instability while symptoms in abduction and external rotation (ER) can indicate anterior instability [32,60,61]. Patients with MDI commonly present with rotator cuff pain, impingement (internal and sub-acromial) and bursitis, which are secondary to glenohumeral instability and poor scapular mechanics [3,62]. 

Patients may report an absence of any mechanism of injury or cause of the onset of their symptoms, or they may recall obvious micro-trauma [58,63]. Microtrauma may include an increase in the load of a particular activity, or similarly a resumption of activity following a significant period of reduced loading [22]. Patients should be questioned regarding any significant history of shoulder trauma (such as a fall and significant subluxation/dislocation) that may indicate structural damage to the shoulder. If the patient reports a significant shoulder trauma in the history, imaging is warranted [64]. MRI is the gold standard when assessing capsular and labral integrity [12]. In cases of atraumatic MDI, MRI is often clear of a significant structural lesion. Blunting of the labrum, bursitis, and rotator cuff tendinopathy may be seen on imaging and are typically the secondary signs of underlying glenohumeral instability [65,66].

Some patients may be able to voluntarily sublux and relocate their shoulders with selective muscle patterning. Volitional instability is a small subgroup of MDI and treatment primarily focuses on education to cease the habit [47].

### 4.2. Physical Examination 

Although there are some discrepancies, the general consensus is that diagnosis is based on a positive test for inferior instability and a positive test for anterior and/or posterior instability [35,45,58,67]. A recent randomised controlled trial adhering to recommendations for diagnosing MDI [9] based their MDI diagnosis on the following criteria: A positive sulcus sign for inferior instability [3,57,59,68].A positive test for at least one direction (anterior and/or posterior), for at least two of three following tests:
Anterior and posterior draw tests in (10°–30°) abduction [57,60,68].Anterior and posterior draw tests in (80°–120°) abduction [57,60].Anterior [3,61] and posterior [62,64], apprehension tests.

These tests are valid and reliable for diagnosing directions of instability when apprehension (including muscle guarding) is used as a criterion for a positive result and not just signs of asymptomatic laxity [9,55]. Caution must be exercised when performing apprehension tests since they (i.e., posterior subluxation for the posterior apprehension test) could be unsafe if performed to their full extent and may risk structural damage to the shoulder. Performing the tests to the onset of symptoms is typically more appropriate. Determining if a test reproduces symptoms of instability has more clinical relevance than grading the degree of humeral head translation, as attempting to grade the level of laxity has been shown to have fair to poor intra-observer reliability [69,70] and the degree of laxity can vary considerably in the normal population [69,71]. 

Apprehension is well documented to be a symptom of glenohumeral joint instability [5,8], but is often associated with structural causes of instability as would occur with a traumatic onset [13]. Passive glenohumeral joint integrity and using apprehension as the criterion for a positive result may result in a false negative in the MDI patient who is unstable actively in functional tasks [30,37,43]. With this in mind, the (i) presence of pain in combination with an aberrant humeral head translation may indicate instability in this population and (ii) assessment of motor control deficits and the “effect of correction” should be included in the MDI screening process [5,72,73,74]. 

### 4.3. Assessing Motor Control Deficits and the Effect of Correction

At a minimum, active flexion and abduction range of motion should be assessed to determine the presence of scapular dyskinesis and aberrant humeral head translations. The therapist will typically observe a lack of scapular upward rotation at rest and through range and possibly some scapular medial rotation (winging) and anterior tilt compared to the unaffected side (or compared to “normal” if both sides are affected) [53]. Range of motion may appear “blocked” due to a motor control deficiency preventing smooth articulation of the humeral head on the glenoid fossa through range. Once the presence of motor control deficits has been established, the effect of correction can be assessed. The effect of manual correction in the assessment of MDI has been previously described [5] and involves 

The therapist choosing an objective test while noting the patient’s faulty scapular and/or humeral head biomechanics during that test in association with symptoms;Applying manual correction of the scapula, humeral head and possibly a combination of both to correct the faulty mechanics while reassessing the test, noting any improvements in range of motion, pain, subluxations/dislocations and/or strength on reassessment.

A positive corrective test should be at least 20° improvement in active range before the onset of symptoms [9]. Scapular correction has been used by numerous authors for assessment of other shoulder pathologies and has good inter-rater reliability [74]. Correction of the humeral head has been validated in anterior instability [75,76] and is still evolving in posterior instability.

Scapular correction is often assessed using active flexion and abduction range of motion by gently guiding the scapula into the position that corrects the faulty motor pattern (typically upward rotation and possibly some posterior tilt) through reassessment of active range (Figure 1). Posterior humeral head translation is commonly assessed with active flexion, or active or isometrically loaded horizontal flexion and corrected with the pad of thumb to limit posterior humeral head translation during reassessment (Figure 2a). Anterior humeral head translation is commonly assessed with active abduction, or active or isometrically loaded ER at 0 or 90° elevation and corrected with four fingers to limit anterior humeral head translation during reassessment (Figure 2b). 

Correction techniques not only determine if there is a motor control pattern component of the patient’s presentation but assist in determining if the patient is likely to respond to rehabilitation and what exercise drills are best suited to commence the program. For example, if a patient’s symptoms improve with a scapular upward rotation correction, then they are likely to improve with rehabilitation that commences with exercises that restore that position of the scapula [7,8,9]. Similarly, if posterior correction of the humeral head improved an objective test, then the rehabilitation program should have a particular focus on activating muscles that limit the posterior humeral head translation [60]. 

Failure to improve with correction could occur for several reasons. First, the therapist may have miscalculated the correction strategy and other corrections may need to be attempted. Second, a significant structural lesion or reactive tendinopathy may limit the effect of correction and further investigations, or rest and reassessment, respectively, may be warranted. Third, severe and long-standing aberrant upper limb motor patterning may result in strong muscle guarding against the clinician’s assistance, rendering the patient uncorrectable on the initial assessment. 

## 5. Treatment

The recommended initial treatment for MDI is a shoulder muscle rehabilitation program [50,77]. The rationale is that improving scapular, rotator cuff and deltoid function compensates for a lack of passive stability and restores active control of the shoulder [32,53,61,78]. In a small sub-group of patients whose acquired lesions result in failure of conservative management, surgery may be considered, and only when faulty mechanics have been largely resolved [1,54,63,79,80]. Early surgical referral may be warranted in an MDI patient who has sustained a significant traumatic structural lesion [55]. Surgery aims to restore passive support by repairing the lesion and reducing capsular volume [81]. 

Five published systematic reviews [13,50,51,52,82], found rehabilitation with a strengthening program has benefits for patients with atraumatic shoulder instability or MDI; however, the quality of the evidence is limited. Issues with much of the current conservative MDI literature include heterogenous MDI patient populations (e.g., some with likely structural lesions and others without), poor reporting of rehabilitation program details, a lack of baseline outcomes and the use of inappropriate outcomes. Failure to grade the level of evidence for outcomes of interest in some systematic reviews [52,82] also leads to confusion for clinicians when interpreting review conclusions.

To date, there are only three programs for atraumatic shoulder instability with enough detail to replicate in the clinical setting [5,6,83,84]. Of these three, only the Watson Instability Program (WIP^1^) has been shown to significantly improve the outcomes of patients with MDI in an RCT [9], providing the highest level of evidence when treating MDI conservatively. 

The Watson Instability Program (WIP^1^) was first published in 2016 and has a focus on normalizing patient-specific faulty scapular motor control, centring the humeral head and progressing exercises into functional and sports-specific ranges. The exercises can be tailored to the severity of the patient’s presentation (i.e., supine or side lie versus standing). In our RCT comparing the WIP^1^ and the Rockwood Instability program for patients with MDI [9], the WIP^1^ had statistically, and clinically significant improvements compared to the Rockwood program on the Melbourne Instability Shoulder Score (MISS) (24 weeks from baseline), the Western Ontario Shoulder Index (WOSI)(12, and 24 weeks from baseline) and pain scores (24 weeks from baseline). We hypothesised that the program’s efficacy may have been due to the progression of exercises into functional ranges [9] and the potential effects that a motor control training program has on the central nervous system [85,86,87]. This theory is currently being tested in a functional MRI study [88]. The efficacy of the WIP^1^ has also been tested in two pre-post intervention studies. One study [8] included 46 patients with MDI and demonstrated that completing 12 weeks of the WIP^1^ significantly improved scapular upward rotation, muscle strength, movement variability and functional outcomes measures. The other study [7] included circus performers with atraumatic shoulder instability and was delivered exclusively via Telehealth. After a 12-week program, performers had significantly improved MISS and WOSI scores at 6, 12, and 24 weeks from baseline, and significantly improved scapular upward rotation and muscle strength at final follow-up (24 weeks). These findings demonstrate that the WIP^1^ can mitigate risk factors (reduced scapular upward rotation and reduced muscle strength) for shoulder pain in active sporting populations and that good treatment effects can still be achieved with the delivery of the program online; improving accessibility for patients who are unable to see a clinician face to face. 

## 6. The WIP^1^ Updates

The WIP^1^ aims to correct faulty scapular and humeral head biomechanics before progressing exercises into functional and/or sport-specific ranges. The program has two components: (1) assessment of correction and (2) treatment. Below is a summary of the stages of the WIP^1^ including the three updates to this previously published program. 

### 6.1. Assessment of Manual Correction 

Assessment of scapular and humeral head correction is a fundamental part of this program as it determines what patient-specific scapular and/or humeral head position(s) the patient will need to train and maintain throughout the program. In addition to the effect of therapist-assisted scapular correction, we have recently added the assessment of patient self-activated scapular correction to determine the effect of active versus passive corrections (*update 1*). After choosing an appropriate active objective test that elicits symptoms (e.g., active flexion) patients are asked to perform 10 to 20 repetitions (+/− 5 s holds) of their specific scapular corrective drill before the objective test is reassessed. Patients who improve their objective test/s with active self-correction (in addition to therapist-assisted correction) may progress at a faster rate or have a greater benefit from the program as they are able to instantly improve their motor control pattern and hence stability through active awareness [89]. Patients who improve only with passive correction are still likely to benefit from the program [9] but may take longer to establish good motor control. 

### 6.2. Treatment 

The treatment component has 6 stages. Most stages have a scapular phase, which the patient must master, before moving on to the arc of motion phase. 


**
*Stage 1*
**



**
*Scapular phase*
**


The aim of the scapula phase of stage 1 is to regain scapular motor control, using the position found best in the assessment. This is the most important phase and usually involves an upward rotation, and elevation drill, possibly with some posterior tilt. Upward rotation of the scapula assists in orienting the glenoid more superiorly to increase the bony contact area with the humeral head. This increases stability, improves the length–tension relationship of surrounding muscles, and assists in reducing the quintessential sulcus sign [31,32,33,34]. This drill is commonly performed in standing, with the arm in 20 to 30° abduction as this position has been shown on fine wire EMG to significantly recruit more of the upward rotators of the scapula compared to the arm by the side (Figure 3) [90]. In patients with rotator cuff pain, this exercise may need to be commenced with the arm by the side until cuff symptoms settle. 

Mirrors, shoulder models, and manual assistance can be utilized to assist the patient in learning the correct movement. Commonly 1–2 cm of elevation, 20° of upward rotation and ½ to 1 cm of posterior tilt achieves an adequate scapular position. Care must be taken to avoid creating any cervical spine symptoms. As MDI commonly presents with concurrent neck pain and weakness [29,61,91], many MDI patients need to perform substantial cervical spine rehabilitation prior to commencing their shoulder rehabilitation. 

A motor control dosage is initially performed for this drill, as the aim is to alter faulty scapular motor patterns. Typically, the patient is asked to achieve 1–2 sets of 20 repetitions (5 s holds), three times a day as there is some evidence that this may assist in central motor reorganization [92]. Once the patient can achieve this drill with the weight of the arm, then a scapular resistance band is added. A scapular resistance band is a TheraBand^TM^ looped around the scapula that is anchored anywhere to adequately resist the patient’s drill. It may be placed on the foot to resist upward rotation and elevation (Figure 4) or anchored more anteriorly to resist upward rotation and posterior tilt. 

Once the patient has achieved the motor control dosage with the band then the additional load can be applied to the drill. Recently, we have found that a TheraBand^TM^ anchored under the foot and held in the hand to provide resistance to the shrug is better tolerated than holding weight as there is less load during the eccentric lowering phase (*update 2*) (Figure 5). Typically, patients work from a yellow (light resistance) to red (medium resistance) then green (heavier resistance) band under the foot. Once the patient can perform 20 repetitions twice a day with a green band under the foot, they usually have enough control of inferior glenohumeral joint motion to cope with a ½ to 1 kg weight in the hand. Patients are also instructed to release their scapular upward rotation short of their complete scapular resting position since the position typically involves excessive scapular downward rotation. When the patient has achieved the motor control dosage with 1 kg in the hand and the scapular resistance band, they have usually established enough scapular motor control to move on to the arc of motion phase [9].


**
*Arc of Motion Phase: Control of the coronal plane 0–45 degrees of elevation*
**


The aim of the arc of motion phase in stage 1 is to regain humeral head control at 0° of elevation in the coronal (abduction) plane. When commencing an arc of motion phase, exercises often commence with a short arc of motion (small range) within mid-range and progress to larger arcs of motion as the patient gains humeral head control through a larger range. Exercises typically commence with light TheraBands^TM^ (yellow or red) and weights (250–500 g) and progress to heavier bands (green/blue) and weights (2 to 5 kg+) depending on the patient’s physical build and exercise goals. 

Exercises in this phase include extension, internal rotation (IR) and ER of the glenohumeral joint. Short arc extension with a TheraBand^TM^ is often commenced first because it is easier for the patient to control. Initially, care must be taken to limit the extension drill to the side of the body as extension past the side of the body can encourage anterior scapular tilt and anterior humeral head translation. Once the patient can control extension, short arc ER can be commenced. Extension and ER drills have a high degree of the posterior deltoid (extension) [93] and infraspinatus and teres minor activity [94], and are thus advantageous for gaining control of the posterior humeral head translation [60]; however, care must be taken when prescribing glenohumeral joint rotation exercises. Prescribing ER too early, prior to establishing good extension and scapular control, can cause symptoms of inferior instability in some patients [95]. In such cases, extension rows may need the building to double TheraBand^TM^ (green and red) and occasionally, extension control with TheraBands^TM^ to 45° and even 90° of elevation (described below) may need to be gained prior to revisiting ER (*update 3*).

IR is typically commenced after extension and ER control is established as IR can cause an excessive anterior tilt of the scapula and pectoralis dominance if commenced too early [96]. IR has a high degree of subscapularis activity [97] which assists in limiting anterior humeral head translation [98]. A humeral headband [5,6,99], can be used with arc of motion exercises which involve placing a TheraBand^TM^ around the patient’s proximal humeral head (Figure 6). The patient generates a small isometric extension force to counteract the flexion force of the band. This has been shown to increase activation of the rotator cuff on EMG, which may enhance glenohumeral joint stability [99]. 

Side lie ER with weight is added to gain additional muscular support posteriorly. This exercise is initially performed off a platform (Figure 7) to limit IR across the body, which can exacerbate the posterior humeral head translation [48,60]. 

For most exercises, the patient continues to use their scapular resistance band. This increases scapular muscle recruitment during IR and ER exercises and so is a useful tool to enhance scapular muscle recruitment [99]. The patient is instructed to “set” their scapula into their corrected position and maintain this, prior to performing their arc of motion exercise. In the early stages of the program, the patient will reset their scapula against the band for every repetition. As scapular motor control improves, the patient will be able to position their scapula into their corrected position and maintain it for the entire set. 


**
*Stage 2: Posterior Muscular Development*
**


The aim of stage 2 is to gain more posterior musculature to further prevent posterior humeral head translation and prepare for flexion-based drills. A large proportion of patients with MDI have a predominance of posterior instability [8,9,100]. Stage 2 aims are achieved by increasing the load of scapula, side lie and ER drills. A bent-over row with weight can be added to strengthen the posterior deltoid. Care must be taken on the release of a bent-over row to ensure that no subluxation occurs with the weight in the hand. A standing row with a TheraBand^TM^ at 45° of elevation is prescribed to commence upward rotation control of the scapula in a higher range (Figure 8). 


**
*Stage 3: Sagittal (Flexion) plane control in 0 to 45 degrees of elevation*
**


The aim of stage 3 is to establish flexion-based control. Flexion is important to introduce as it is functional and has a high degree of serratus anterior activation [101] which is an important muscle for controlling scapular upward rotation. Flexion drills are typically commenced as a punching motion with a TheraBand^TM^ anchored behind the patient (Figure 9). In patients with a high degree of posterior instability, the forward motion may need to be commenced in the scapular plane (from light to heavier bands) and progressed around into the sagittal plane as the patient gains control. If the patient has difficulty controlling flexion-based drills, they are regressed to stage 2 for more posterior muscular strengthening. 


**
*Stage 4: Sagittal and Coronal plane control in 45 to 90 degrees of elevation*
**



**
*Scapula phase*
**


The aim of the scapula phase in stage 4 is to gain scapular control at 90° of elevation. This is achieved by progressing the standing row at 45° of elevation (Stage 2) up to 90° of elevation with a red then green TheraBand^TM^. 


**
*Arc of motion phase*
**


Once scapular control is established with the high row at 90° of elevation, short arc ER then IR are added to the motion. ER at 90° elevation in the coronal plane can cause both excessive anterior and/or posterior translation, depending on the patient’s primary direction of instability and may need to be commenced in the scapula plane (from light to heavier Therabands^TM^) then progressed back into the coronal plane as the patient gains humeral head control. ER at 90° encourages a strong activation of the posterior cuff to limit the posterior humeral head translation [60]. Once ER control in the coronal plane has been established, IR can be commenced. 

Gaining control of flexion and horizontal flexion at 90° of elevation and above is important for asymptomatic daily activities and can be particularly challenging for patients with a predominance of posterior instability. If control of these positions is not established, patients can have ongoing symptoms. To gain control across the body, ER at 90° elevation is performed in increasing increments (over several weeks) from the coronal plane to the sagittal plane using TheraBands^TM^ (Figure 10a) and weights on a support (Figure 10b). Once in the plane, flexion-based drills with a TheraBand^TM^ then weights are also progressed into higher ranges of motion.


**
*Stage 5: Deltoid Function*
**


The aim of stage 5 is to develop specific muscular strength in all three parts of the deltoid. Although deltoid work has commenced in previous stages, stage 5 aims to build more muscular strength through a variety of ranges. The deltoid function is inhibited in MDI [37] and in addition to the rotator cuff, is important for stabilizing the humeral head [102]. The posterior deltoid is strengthened with a bent-over row at lower ranges, progressing to higher ranges of elevation (Figure 11a). The anterior deltoid is strengthened with standing flexion bands, supine push presses (taking care that no posterior instability occurs with a push) and controlled overhead presses (Figure 11b). The middle deltoid is strengthened with short lever lateral raises (Figure 11c). 


**
*Stage 6: Sports Specific and Functional Stage*
**


The aim of stage 6 is to develop sports specific and functional specific control where the exercises are tailored to the needs and goals of the patient. For this stage, a sports specific drill is often broken down into parts and practised. For example, the pull and catch phase of a swim stroke may be practised separately (part practise) until being placed together in the pool (full practise). The same principles of the program apply when commencing a new sport specific exercise; ensure scapular position is maintained, gain control of the humeral head motion (short arc to larger arc) and progressing from light to heavier loads. Dosages in this stage need to mimic what is functionally required by the patient. 

The kinetic chain is important to address, especially in the sporting patient [103,104,105]. Lower limb kinetic chain exercises can commence early in the program while the shoulder is regaining function, and more integrated, whole-body motions (specific to the patient’s sport or occupation) can be implemented once the shoulder is stable enough to contribute to the sequence of movements. Careful management of load (both volume and duration) must be considered with return to full sporting and/or occupation participation. Tools to reduce loads should be implemented in the initial return phases such as fins for swimming and alternating training and rest days. 

### 6.3. Return to Sport and Detecting Change with Treatment

MDI patients who wish to return to high-level activities should pass a strength test prior to returning to training and then the full competition. In a group of patients who had undergone a Bankart repair for anterior shoulder instability, Drummond et al., [106] found those who passed a strength test on an isokinetic machine (with a pass being 90% strength of the unaffected side) were nearly five times less likely to have a recurrent instability event compared to those who returned to sport based on the time from surgery alone. Although this study was in a post-surgical population, the principles of ensuring adequate scapular and glenohumeral muscle strength for safe return to high-level activity still apply to a non-traumatic population. Previous MDI studies [7,8,9] have shown that group improvements in functional outcomes (including sporting and activity participation) are related to group improvements in muscle strength on a hand-held dynamometer. Shoulder strength testing with hand-held dynamometry has good inter and intra-rater reliability when performed by an experienced practitioner and the protocol used in MDI clinical trials has been published in detail [107]. Given that a significant proportion of MDI patients may have bilateral problems, normative data in aged-matched sporting populations needs to be established in future research studies. 

Detecting functional change with treatment should be completed with instability-specific outcome measures. The MISS and WOSI are sensitive and specific for measuring changes in the shoulder instability population and have a high responsiveness [108]. 

### 6.4. Dosage, Timelines and Discharge

Most exercises commence with a motor control dosage (1–2 sets of 20 repetitions, 5 s holds, 2–3 times a day), then build to an endurance dosage (2 × 15 repetitions, 2–3 s hold, 1× day). Exercises with weights progress to a strength (3 × 10^–12^ repetitions every 2nd day) and possibly hypertrophy dosage (3 × 6^−8^ repetitions, 3× week) if required for sports [109]. 

Once patients have completed the program, they are encouraged to continue a maintenance schedule two to three times a week, which involves a combination of band and weight exercises targeted to maintain the control of the patient’s primary direction of instability and their functional requirements.

Patients are likely to have a significant improvement in functional outcomes after 6 weeks of the WIP^1^ program though scores of 80% of a normal shoulder are seen closer to 12 weeks [7,8,9]. The largest treatment effects are seen at 24 weeks post rehabilitation [9], which is when most overhead sporting patients may expect to recommence training. Timelines for recovery may extend in the case of concomitant cervical spine pain, postural thoracic outlet syndrome (TOS) and connective tissue disorders. Surgery may be considered in patients who fail 6 to 9 months of evidence-based rehabilitation [9,25,32,61,110].

### 6.5. Associated Factors That Alter the Focus of Rehabilitation 

While the clinician should assess each individual MDI patient for their aetiology sub-group [48] and contributing factors to their pathology; key associated factors may alter exercise selection, course of management and recovery timelines. MDI patients with connective tissue disorders typically require modification of their rehabilitation loads, exercise drills and rates of progression due to the fragility of their tissues [95]. A small subgroup of MDI patients with severe and long-standing aberrant motor patterning may not respond to correction, at least initially. Given the cortical changes associated with atraumatic shoulder instability [111], it is plausible that these patients may have central changes that need addressing through brain training prior to commencing low-level scapula drills. MDI patients with significant traumatic or micro-trauma structural lesions may require surgical stabilisation to restore the passive integrity of the shoulder if evidence-based conservative management fails [48,51]. MDI patients with concurrent cervical spine symptoms and/or postural TOS [91,112] typically need to normalise deep cervical flexors (DCF) weakness prior to prescribing standing scapular drills as performing scapular upward rotation and elevation on a weak and painful neck are likely to exacerbate neck symptoms. Table 1 outlines these key associated factors, their typical presentation, and adjustment to management. As with any patient, identification of any psychological or systemic disorders warrants referral and multidisciplinary management with the appropriate professional.

## 7. Conclusions

The aetiology of MDI is multifactorial, and individual contributing factors need to be considered in management. Conservative management is the first line of treatment in MDI, with the WIP^1^ having the current highest level of evidence for effectiveness. Instability-specific outcomes and strength tests can be used to measure the outcome of treatments. 

## Figures and Tables

**Figure 1 jcm-11-05140-f001:**
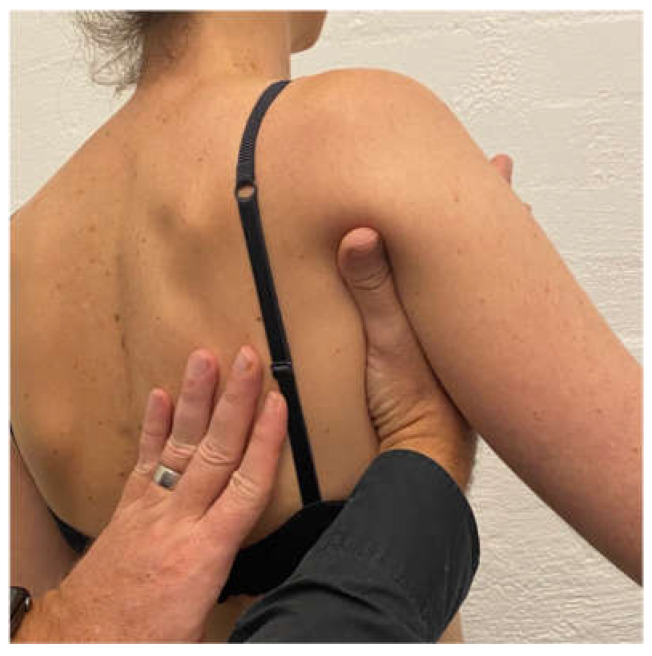
Manual correction of the scapula into upward rotation.

**Figure 2 jcm-11-05140-f002:**
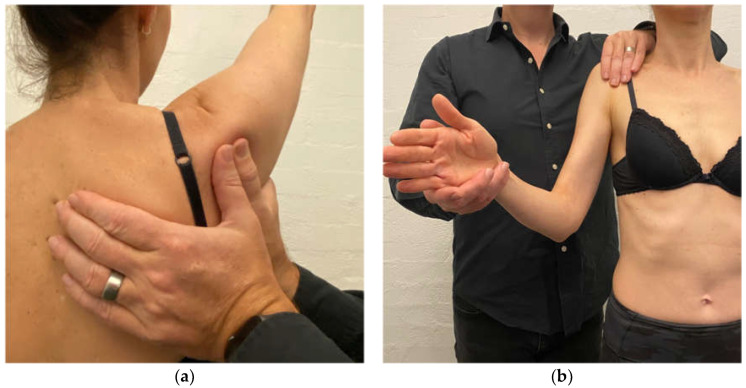
(**a**) Posterior humeral head correction during active shoulder flexion; (**b**) anterior humeral head correction during isometric load of external rotation.

**Figure 3 jcm-11-05140-f003:**
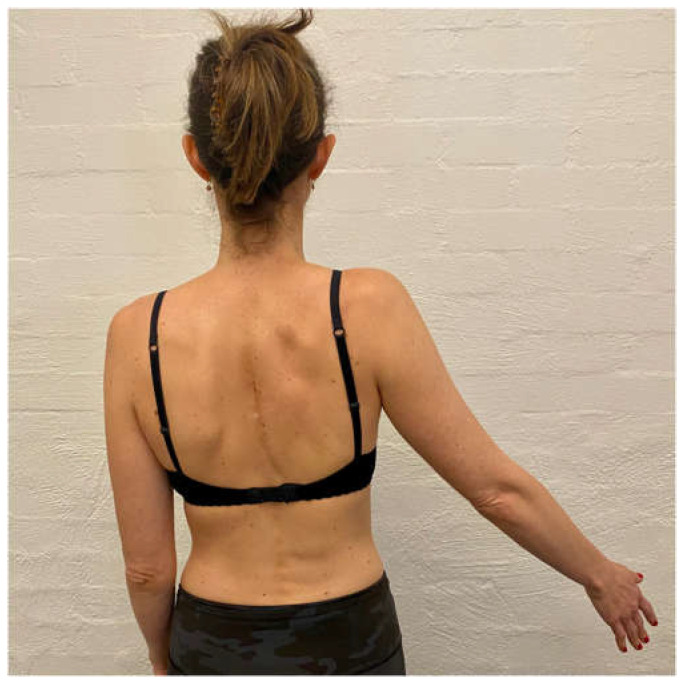
Upward rotation/elevation drill with the arm in 20 to 30° abduction.

**Figure 4 jcm-11-05140-f004:**
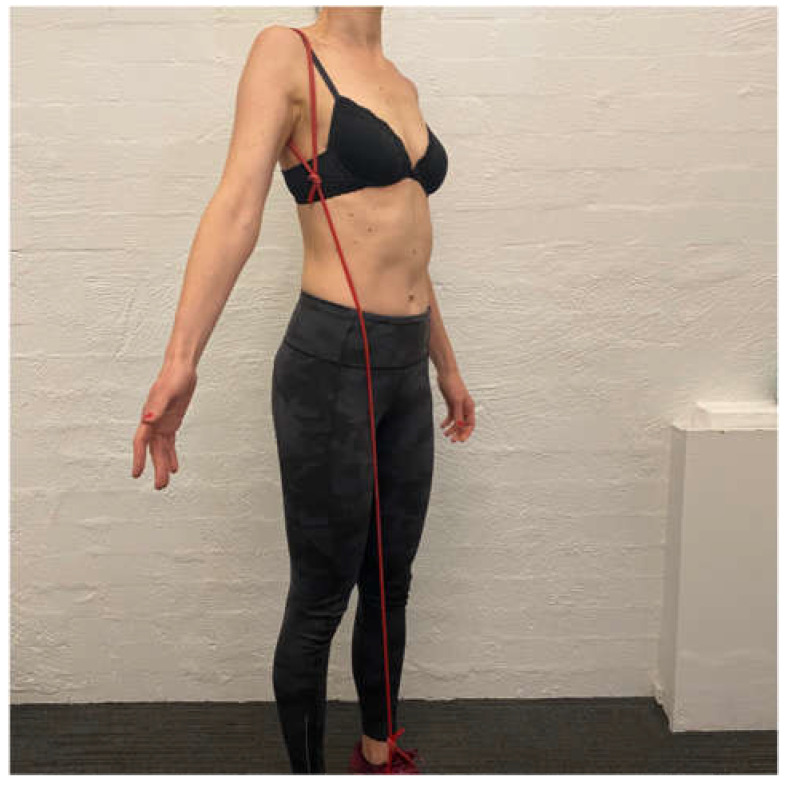
Upward rotation/elevation drill with scapular resistance band anchored to the foot.

**Figure 5 jcm-11-05140-f005:**
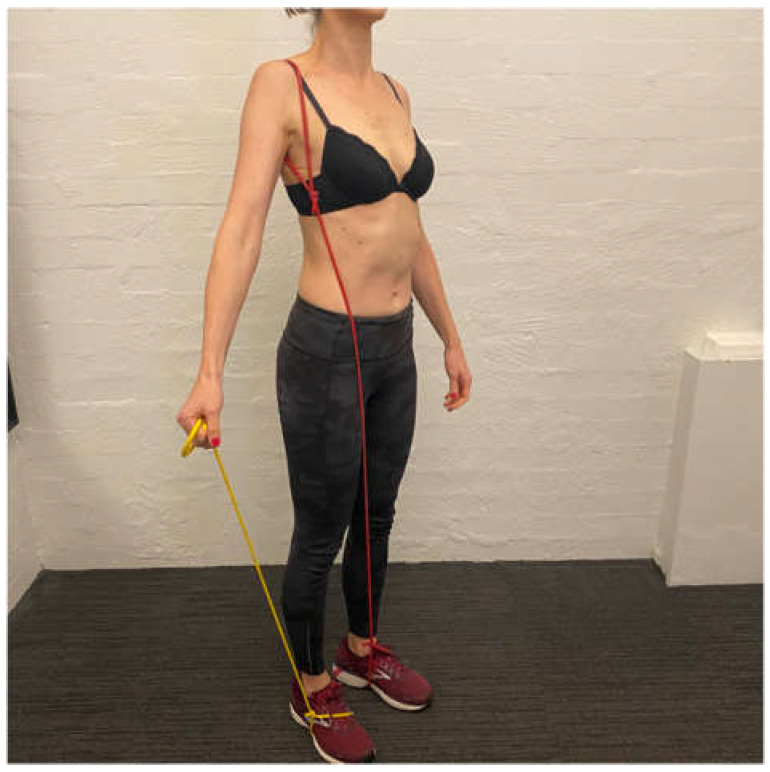
Resistance band under foot during scapula upward rotation/elevation drill.

**Figure 6 jcm-11-05140-f006:**
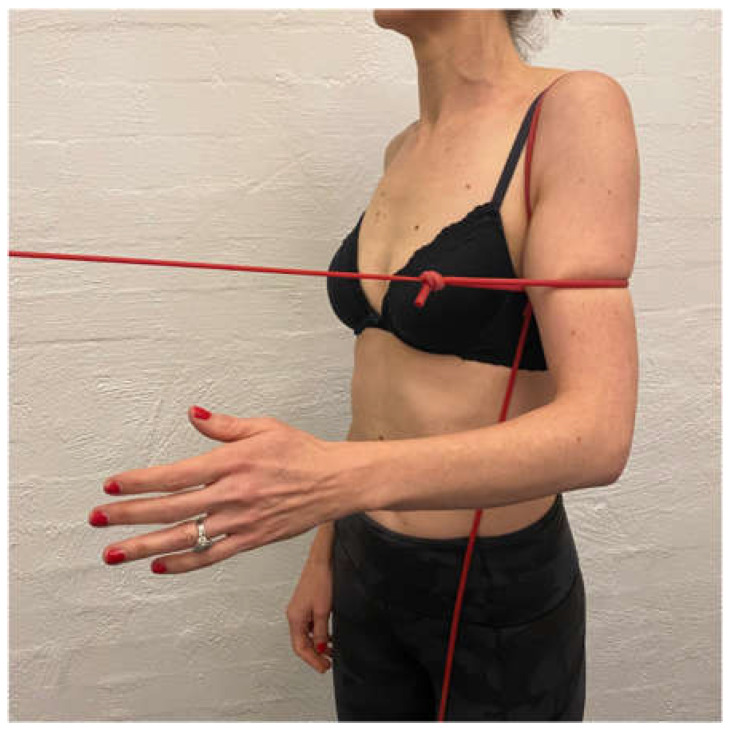
Humeral head band.

**Figure 7 jcm-11-05140-f007:**
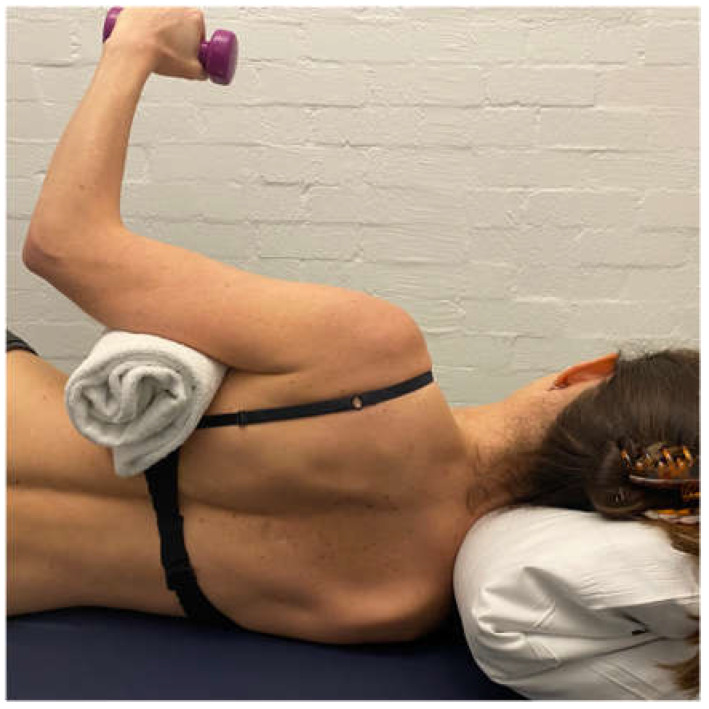
Side-lie external rotation.

**Figure 8 jcm-11-05140-f008:**
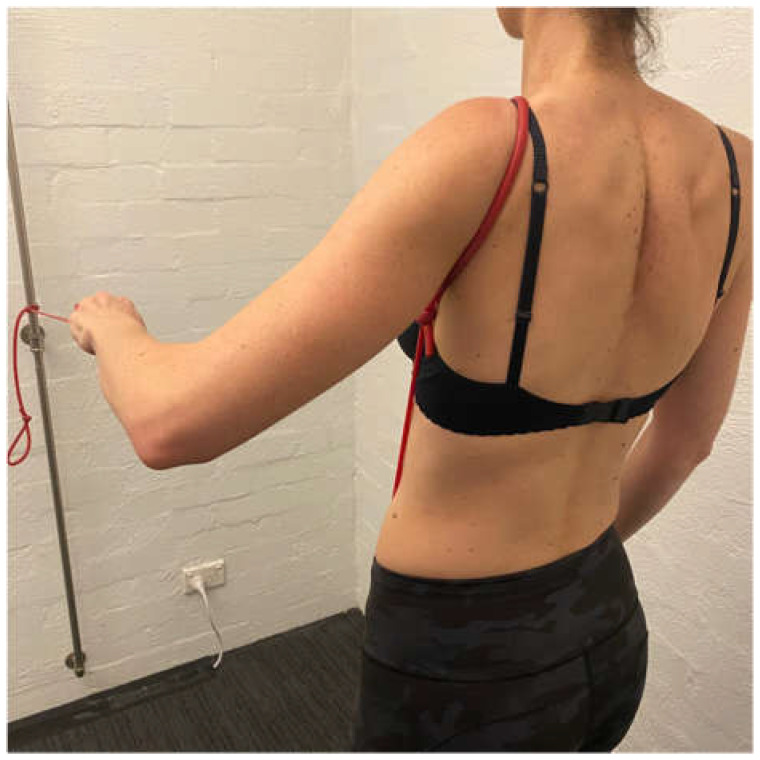
Row at 45 degrees of shoulder elevation.

**Figure 9 jcm-11-05140-f009:**
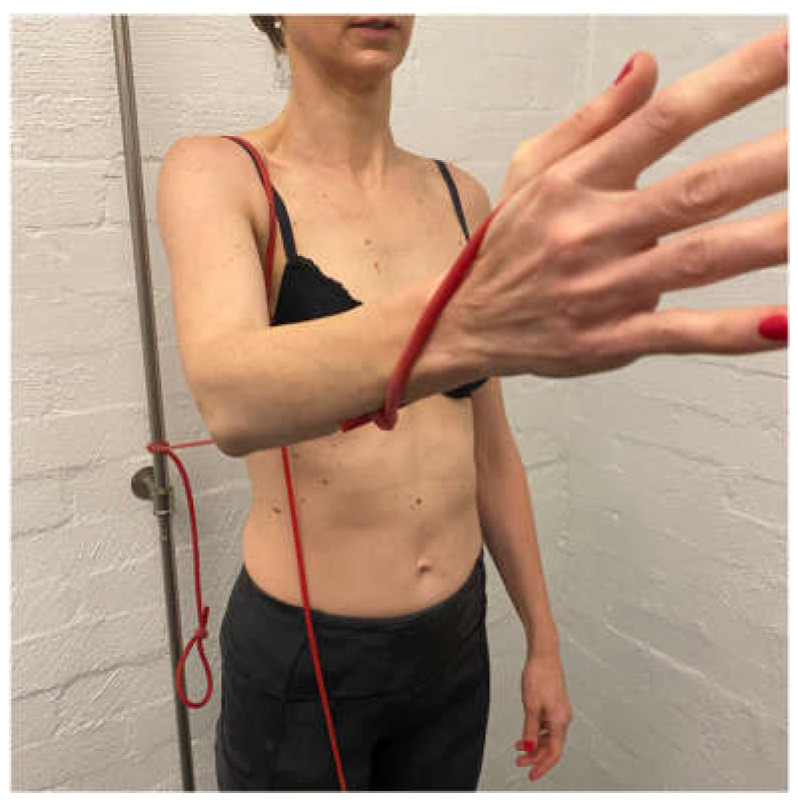
Theraband^TM^ flexion drill.

**Figure 10 jcm-11-05140-f010:**
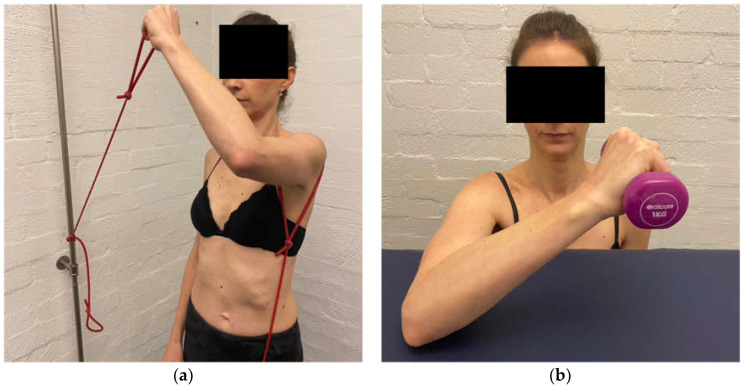
(**a**) External rotation in the sagittal plane with TheraBand^TM^. (**b**) External rotation in the sagittal plane with weight.

**Figure 11 jcm-11-05140-f011:**
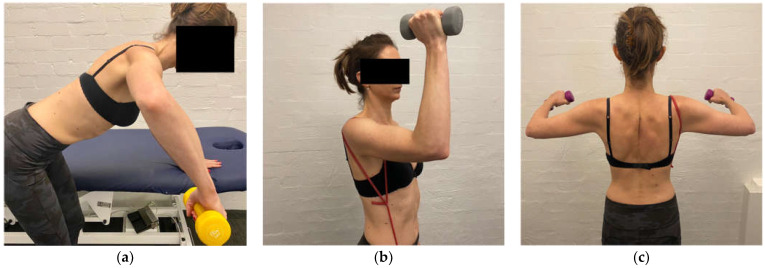
(**a**) Bent-over row at 90° of elevation (**b**) overhead press (**c**) short lever lateral raise.

**Figure 12 jcm-11-05140-f012:**
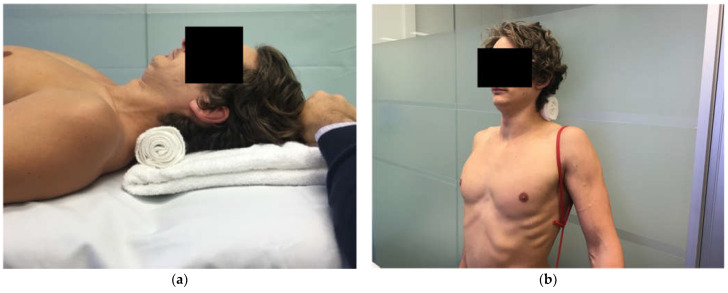
(**a**) Supine deep cervical flexor exercise (**b**) standing deep cervical flexor exercise.

**Table 1 jcm-11-05140-t001:** Associated Factors that Impact the Focus of Rehabilitation.

Associated Factor	Clinical Presentation	Key Modifications/Explanation and Examples
MDI with connective tissue disorders	Examples: Hypermobility syndrome Marfan’s syndrome Ehlers–Danlos syndrome May have concurrent musculoskeletal disorders (e.g., craniocervical instability, cervical spine weakness, patella-femoral joint pain) May have concurrent systematic disorders (e.g., heart murmur, poor gastro-intestinal mobility, valvular heart disease)	Key modifications: Rehabilitation loads, exercise drills and rates of progression need to be modified due to fragility of tissues. Explanation and Examples: Exercise loads may need to be applied and progressed with TheraBand^TM^ instead of weights for a long period. If weights prescribed, may need to limit to a maximum amount (e.g., 2–3 kg for bent over row, 2–3 kg supine flexion drills, 1–2 kg overhead press). Drills rarely performed to the very end ranges of motion. Entry into the rehabilitation program is typically lower (e.g., supine or side lie scapular drills instead of standing). Progression through the program is slower. Focus on 1 to 3 motor control drills with light load for several weeks before progression. May need to be counselled on realistic expectations of their shoulders and bodies regarding activities (e.g., contact and throwing sports unlikely and/or high risk).
MDI with severe motor patterning and unamenable to correction	Patient *does not* respond to scapular correction (at least initially) Typically, a long history of symptoms May have psychosocial involvement Manual correction typically met with severe and involuntary muscle guarding which may appear as glenohumeral capsular stiffness. *Note*: True glenohumeral joint stiffness is unlikely in the absence of trauma and in the typical MDI age range [24,113]. Possible central involvement [111]. Patients may have poor laterality (left and right recognition) [114]. When asked to perform imagery of simple shoulder motion (e.g., flexion, abduction, ER) patients may report ‘gaps’ in imaged motion (e.g., missing pieces of flexion range), or an increase in pain during attempted imagery [115,116].	Key modifications: Can still respond well to the rehabilitation program; however, commence the program at a lower level and progress at a much slower rate. Brain training with graded motor imagery (GMI) may be required in a small sub- group of patients. Explanation and Examples: Scapular drills may need to be performed in side-lying, supine or isometrically prior to performing the drill isotonically. Fewer repetitions of the exercise may be required initially. Once patients can progress scapular drills to standing, then they can usually progress along the standard WIP^1^ guidelines. Once these patients have regained a sense of normal scapular motor patterning, it is not uncommon for them to become correctable. A small sub-group of patients may be unable to perform the simplest of scapular drills (e.g., supine) and may require brain training with GMI prior to performing active motor drills. GMI has good evidence for efficacy in musculoskeletal disorders [117,118]. Normalise left and right recognition if abnormal [114,117,119]. Imagery training such as mental rehearsal of a correct scapular motor pattern (often facilitated by watching a video of “normal” scapula motion), simple shoulder range of motion, or shoulder-based activities of daily living (e.g., washing hair, reaching up into a cupboard, getting dressed) within pain limits [117,119]. Mirror therapy is rarely required in this context though may be implemented if deemed necessary [117,119,120].
MDI with a history of microtrauma or trauma	History of significant trauma (e.g., fall and contact with an external object and conscious awareness of the onset of pain and/or subluxation/dislocation). Or Microtrauma (gradual or acute overload of muscular, e.g., increase in volume of tennis with a heavier than usual racquet). More likely to have 2 (anterior and/or posterior) rather than 3 directions of instability Increased likelihood of structural lesions	Key modifications: Microtraumatic MDI: If 6 months of evidence-based rehabilitation fails, surgery may be considered. MDI with trauma: rehabilitation to normalise poor motor patters. Early surgical referral warranted in cases of significant traumatic structural lesions. Explanation and Examples Clinician should determine if the patient is amenable to scapular +/− humeral head correction. Correctable = rehabilitation should be commenced to correct any movement errors. If the patient continues to improve over 12 to 24 weeks, then rehabilitation is continued. Non-correctable +/− aberrant motor patterns= commence rehabilitation to determine if the patient’s assessment asterisk/s improve over several weeks and to normalise faulty motor control. If improved over 12 to 24 weeks, then rehabilitation is continued. If not showing any improvement over 2 to 3 weeks of rehabilitation\high quality imaging should be sought. If significant structural lesion present and not responding to conservative rehabilitation, then surgery to restore the soft tissue structures may be indicated [13,51]. Post-surgery, the focus of rehabilitation is still on establishing good scapular and humeral head control, initially at rest and then through movement [5,6,8,9]. Passive stretching of the glenohumeral joint is usually avoided, as most MDI patients will regain their range with gentle active motion. Post-surgical rehabilitation typically has a longer time frame (9 to 12 months) compared to other post-surgical rehabilitation programs for other types of shoulder instability.
MDI with neck dysfunction	MDI patient with concurrent cervical spine pain and/or weakness and/or postural thoracic outlet syndrome (TOS). Typically due to traction of cervical structures and the brachial plexus from a depressed and downwardly rotated scapula [32,91,112] and/or primary cervical spine weakness. Deep cervical flexors (DCF) are weak on pressure biofeedback assessment and/or lack smooth motor control of the neck flexion. Scapular corrective techniques and reassessment of cervical range of motion can assist in determining if the scapula position is a significant driver of neck and/or TOS symptoms.	Key modifications: Normalise cervical spine strength prior to commencing standing scapula drills. Explanation and Examples: Treatment involves offloading the cervical structures with scapular taping and bracing and addressing ergonomics (e.g., arm rest at the desk to prevent excessive cervical traction). DCF activity and strength needs to be normalised as a precursor to prescribing standing scapula drills as performing upward rotation, elevation drills on a weak and painful neck are likely to exacerbate neck symptoms. Drills are then progressed from supine (Figure 12a) to standing against a wall (Figure 12b). Once the patient can perform DCF drills in standing then they are usually ready to add in a standing scapular drill to their DCF drill. Cervical extension drills are also added.

Note: DCF = deep cervical flexors, MDI = multidirectional instability, TOS = thoracic outlet syndrome.

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
