# Peer review of "Advances in the Non-Operative Management of Multidirectional Instability of the Glenohumeral Joint"

_jcm, 2022, doi:10.3390/jcm11175140_

Round 1

Reviewer 1 Report

A brief summary

The aim of this paper was to provide a new approach to MDI non-operative management. The authors have successfully presented a large amount of important information regarding the treatment for MDI, through the shoulder muscle rehabilitation program, including valuable information about the WIP updates.

General concept comments

From the aetiology and classification to clinical presentation and treatment, the paper is appropriate and clearly describes all the data. The pictures used for the evaluation and rehabilitation program are adequate and helpful.

The manuscript is highly relevant to the field of orthopaedic rehabilitation.

Still, less than half of the cited references are published within the last five years. I also suggest reviewing the reference list and completing the date of publishing for a few articles where the year is missing.

Reviewer 2 Report

The paper is a reminder of authors papers from prevoius years.Is not presented a group of patients,with different etiology for MDI,and compare the results according the scale presented after 12,18 and 24 weeks.The results is not relevant if that groups are not made.Is jus a presentation of the procedures.

And in the bibliography should have at least 50 procents after 2015 and is not the case in this paper.
